# Biomass Partitioning and Morphoanatomical Traits of Six *Gymnocalycium* (Cactaceae) Species Occurring along a Precipitation Gradient

**Solana B. Perotti** [1,†]**, Nayla L. Aliscioni** [2,†]**, Natalia E. Delbón** [2,†]**, Mario Perea** [3]**, Ariadna Hammann** [1] **and Diego E. Gurvich** [2,*]

1   Facultad de Ciencias Exactas y Naturales, Universidad Nacional de Catamarca, Catamarca K4700, Argentina
2   Instituto Multidisciplinario de Biología Vegetal, Facultad de Ciencias Exactas, Físicas y Naturales, CONICET-Universidad Nacional de Córdoba, Córdoba B5000, Argentina
3   Departamento de Biología, CEVIR, Centro de Estudios de Especies Vegetales de Interés Regional, Facultad de Ciencias Exactas y Naturales, UNCa, Catamarca K4700, Argentina
*   Correspondence: degurvich@unc.edu.ar
†   These authors contributed equally to this work.

**Abstract:** As a group, cacti are regarded as plants that tolerate water scarcity, since they present a number of adaptations. However, little is known about how species of the family varied their morphoanatomical characteristics along environmental gradients. The aim of this study was to analyze how six *Gymnocalycium* species occurring in three sites along a precipitation gradient (arid site: *G. pugionacanthum, G. marianae*; semiarid site: *G. hybopleurum, G. stellatum;* subhumid site: *G. oenanthenum, G. baldianum*) differ in their biomass partitioning and morphoanatomical characteristics. We collected mature individuals of each species and analyzed their biomass partitioning (to spines, aboveground stem, underground stem, main root, and lateral and thin roots), morphological characteristics (such as size ratios, spine length and width, and areole density) and anatomical characteristics (stoma number, and cuticle, epidermis, and hypodermis width). Species differed, both qualitatively and quantitatively, in most of the analyzed variables. For example, biomass allocated to spines was highest in *G. pugionacanthum*, lowest in *G. baldianum*, and intermediate in the remaining species. However, these variations were not clearly associated with aridity, but were related to the subgenus of the species. These patterns were clearly observed in the PCA. Phylogenetic relatedness is the main factor associated with morphoanatomical characteristics.

**Keywords:** plant strategies; functional traits; succulence; Catamarca province (Argentina)

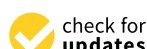



## 1. Introduction

Succulence is a plant characteristic that has evolved numerous times as a consequence of the planet aridification during the Cenozoic era [1]. Functionally, succulent plants have been viewed as a low growing homogeneous group that occurs in stressful environments [2,3]. However, important intragroup variations exist in terms of life forms that surely include other functional differences [4]. For example, in the Cactaceae family, species have evolved in numerous life forms, such as globose, barrel, columnar, epiphytes, and geophytes [5]. Even within a growth form, species may differ in their characteristics since they present very contrasting distributions in terms of environmental characteristics. For example, in South America globose cacti are present from the humid regions of southern Brazil to the hyperarid desert of Chile and Peru [5].

In terrestrial plants, a number of traits were identified as indicators of resource use [3,6]. For example, specific leaf area and wood density were pointed out as key traits, since they are good predictors of a species' response to the environment [7,8]. These traits cannot be measured in globose cactus species since they do not present leaves or a typical

wood. To our knowledge, no study has attempted to identify which traits could be of ecological significance in this group. Particularly, studying species and their traits occurring along a precipitation gradient could give clues about which trait, or trait combination, responds to aridity [9]. We hypothesized that biomass partitioning, and morphological and anatomical characteristics could be important traits in differentiating species present in contrasting habitats.

Cacti exhibit a wide variety of adaptive morphological, anatomical and phylosiological traits that allow them to grow and reproduce in water-limited environments. The different combinations of these traits are specific to species or genera, being of major taxonomic importance; these combinations also vary according to the specific environmental conditions under which species occur [10]. Most cactus species have spines, which have many functions [11]. Spines represent a construction cost to plants; thus, the biomass allocated to spines could be an important characteristic related to the environment [12]. For example, as the environment gets more arid, species could allocate more biomass to spines to improve thermoregulation or to capture more water [11,13]. Many globose cacti present underground organs (both stems and roots) that act as water and nutrient reservoirs [14]. It could be expected that more biomass allocated to underground organs would be related to more stressful environments, like more arid ones [15].

The anatomical characteristics of cactus stems are well described for the common genera from North America. Most of the studies are descriptive, with implications for taxonomy and systematics, or phylogeny and evolution, while a few studies have attempted to relate anatomical characters to aridity [10,16–22]. So far, no anatomical traits of a species group have been studied in relation to environmental gradients. Some of the most important features are those observed in the dermal system, composed of the epidermis and the subjacent hypodermis [10,16,17,23]. The system plays several roles: it supports the stem, helps conserve water, protects internal tissues against sunlight, and provides a defense against pathogenic organisms [19,23]; in addition, it is expected to vary in response to aridity, especially in environments with low availability of water and nutrients, and high solar radiation and temperature.

*Gymnocalycium* is a cactus genus native to southern South America [24,25] that comprises about 50 species. The highest richness is found in the mountains of north-western Argentina, a very heterogeneous region in terms of climatic characteristics, with wet to very dry ecosystems being present [14,24,26]. This genus offers a model to test how environmental characteristics are related to morphoanatomical traits and biomass partitioning. The aim of this study was to analyze how six cactus species occurring along a precipitation gradient (arid site: *G. pugionacanthum, G. marianae*; semiarid site: *G. hybopleurum, G. stellatum*; subhumid site: *G. oenanthenum, G. baldianum*), differ in terms of biomass partitioning and morphoanatomical characteristics.

## 2. Materials and Methods

### 2.1. Study Species and Area

The study was conducted along a precipitation gradient in the Catamarca province, Argentina. We collected individuals of six endemic species of *Gymnocalycium* (Subfamily: Cactoideae, Tribe: Trichocereeae), in three sites (arid, semiarid, and humid sites) in February 2022 (Figure 1). The arid site is in Aconquija, which is in the Monte ecoregion (mean annual precipitation of 380 mm and mean temperature of 16.3 °C [26]), and where *G. pugionacanthum* Backeb. ex H. Till (subgenus Scabrosemineum) and *G. marianae* Perea, O. Ferrari, Las Peñas & R. Kiesling (subgenus Gymnocalycium) occur. The other two sites are in the Chaco Serrano ecoregion; one is a semiarid site near San Fernando del Valle de Catamarca city (mean annual precipitation of 460 mm and mean annual temperature of 19.7 °C), where *G. stellatum* Speg. (subgenus Trichomosemiuneum) and *G. hybopleurum* (K. Schum.) Backeb. (subgenus Scabrosemineum) are present, and the other is a subhumid site in El Rodeo (500 mm and 17.4 °C, of mean annual precipitation and temperature, respectively) where *G. oenanthemum* Backeb. (subgenus Scabrosemineum) and *G. baldianum*

(Speg.) Speg. (subgenus Gymnocalycium) occur. We collected five mature individuals, between 6 and 10 cm in diameter, of each species per site. Two individuals of *G. marianae* died after being collected.

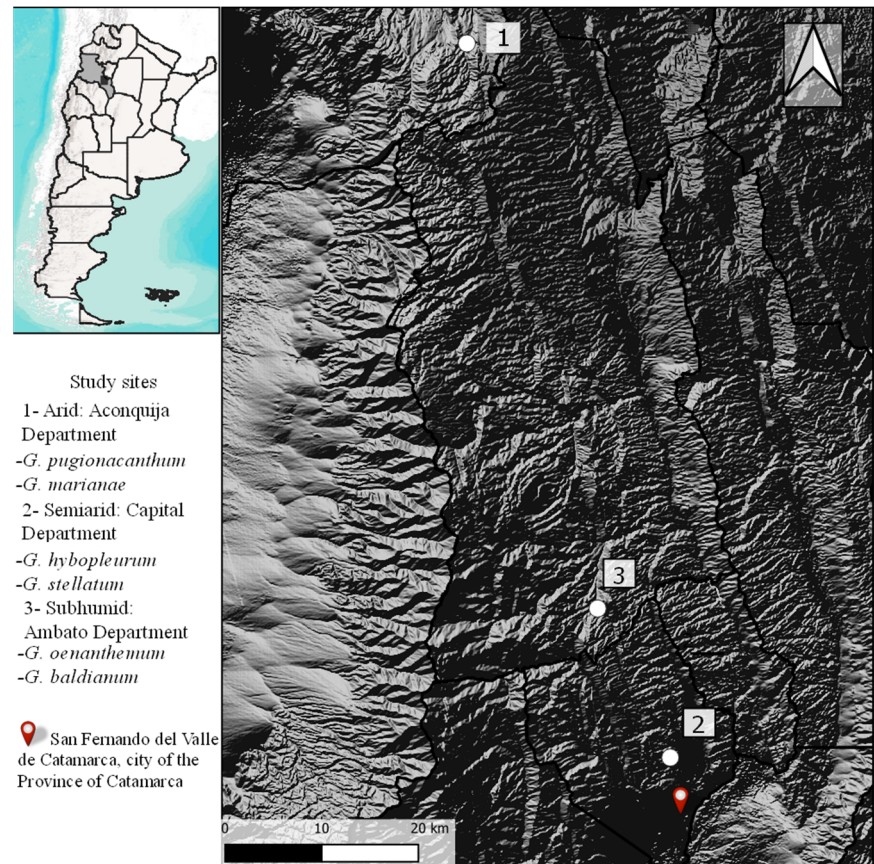

**Figure 1.** Location of the three collection sites of six species of *Gymnocalycium* along a precipitation gradient in the province of Catamarca, Argentina. Numbers/circles indicate the location of the study sites. 1—Aconquija department (*G. pugionacanthum, G. marianae*), 2—Capital department (*G. hybopleurum, G. stellatum*), 3—Ambato department (*G. oenanthemum, G. baldianum*).

## *2.2. Morphological Traits*

We measured the length of each part separately (above and underground stem, and root) of each collected individual. Since the measured characteristics are dependent on the size of the individuals, we relativized the values to facilitate comparison between species. We also counted the number of ribs, calculated the density of areoles in a 9-cm$^2$ area in two different points of the individual, counted the number of spines per areole in two areoles, and measured the length and width in the base of the central or most prominent spine in one areola.

## *2.3. Biomass Partitioning*

We divided each individual into five parts: areoles with spines, aboveground stem, underground stem, main root, and secondary and thin roots. We weighed each section to obtain the fresh weight (FW) and then dried them in an oven at 80 °C until they were totally dry (this process depends on the size and water content of the plant part). After the drying process, we weighed each part again to obtain the dry weight (DW). As mentioned above, these values were relativized for comparisons.

### 2.4. Anatomical Traits

We made temporary histological preparations for light microscope observations. We extracted two surface pieces of approximately 1 cm$^2$, from the middle region of aboveground stem from each individual, and we fixed them with FAA. The fixed material was used to perform temporary slides of epidermis in surface view, using the techniques of peeling or scraping, and stained with Basic Fuchsin and mounted in 50% glycerin [27]. We counted the number of stomata in 1 mm$^2$ in five randomly chosen squares, that is five squares per individual per species. In addition, we made temporary histological slides of the stem in the cross section. The sections were made freehand, stained with astral blue and basic fuchsin, and mounted in 50% glycerin [27,28]. We measured the thickness of the cuticle, epidermis, and hypodermis in 10 different parts of each slide using a micrometric rule. With this information, we calculated the mean value per individual. In addition, we performed additional staining with safranin and phloroglucinol to try to know the chemical nature of the hypodermis.

### 2.5. Statistical Analysis

We calculated the mean and standard deviation for all of the quantitative variables and compared the species with univariate ANOVA tests. A principal components analysis (PCA) was run using FactoMineR and Factoextra packages to understand the main axes of variation and how they relate to environmental characteristics. The first three PC axes were chosen, which accounted for the highest percentage of variation. All analyses and graphics were done using R software in version 4.0.2 and RStudio [29].

## 3. Results

### 3.1. Morphological Traits

We found significant differences among species in most of the analyzed traits. All results are summarized in Table 1. In the most arid extreme of the gradient, *G. pugionacanthum* (Figure 2A) presented the roughest, longest, and widest spines of all the studied species, with the largest underground stem in relation to the total length of individuals. Furthermore, *G. marianae* (Figure 2B) presented a high density of areoles and a high number of spines; however, spines were shorter and narrower than those of *G. pugionacanthum*.

**Table 1.** Summary of all measured variables with mean and standard deviation (SD) for each species. * Significant differences ($p \leq 0.05$). Means with the same letter are not significantly different.

| | Arid | | Semiarid | | Subhumid | | |
|---|---|---|---|---|---|---|---|
| **Variables** | *G. pugionacanthum* | *G. marianae* | *G. hybopleurum* | *G. stellatum* | *G. oenanthemum* | *G. baldianum* | **ANOVA** |
| **Morphological** | **Mean** | **Mean** | **Mean** | **Mean** | **Mean** | **Mean** | ***p*-Value** |
| Length of aboveground stem (%) | 24.98 AB (8.79) | 32.97 B (4.12) | 21.30 A (5.01) | 15.58 A (2.82) | 18.96 A (4.84) | 19.44 A (4.82) | 0.0005 * |
| Length of underground stem (%) | 39.55 B (6.01) | 19.55 A (1.05) | 29.83 AB (5.27) | 32.15 AB (5.30) | 20.09 A (9.66) | 27.41 AB (9.05) | 0.002 * |
| Length of main root (%) | 35.49 A (3.59) | 47.48 AB (3.15) | 48.87 AB (9.05) | 52.26 B (3.68) | 60.96 B (12.63) | 53.15 B (8.15) | 0.001 * |
| Number of ribs | 10 AB (1.23) | 12.3 B (0.58) | 11.8 B (1.64) | 12.4 B (1.82) | 10.4 AB (0.55) | 7.8 A (0.84) | 0.0001 * |
| Density of areoles number/cm$^2$ | 0.36 A (0.03) | 0.81 BC (0.36) | 0.47 AB (0.03) | 0.96 C (0.17) | 0.36 A (0.05) | 1.16 C (0.3) | <0.0001 * |

**Table 1.** *Cont.*

| | Arid | | Semiarid | | Subhumid | | |
|---|---|---|---|---|---|---|---|
| **Variables** | G. pugionacanthum | G. marianae | G. hybopleurum | G. stellatum | G. oenanthemum | G. baldianum | **ANOVA** |
| **Morphological** | **Mean** | **Mean** | **Mean** | **Mean** | **Mean** | **Mean** | ***p*-Value** |
| Number of spines per areola | 5.6 [AB] (1.34) | 9 [C] (1.73) | 7.4 [BC] (1.67) | 3.8 [A] (0.84) | 7.4 [BC] (0.55) | 5.8 [AB] (0.84) | 0.0001 * |
| Spine length (mm) | 27.05 [C] (2.99) | 12.52 [AB] (2.29) | 20.81 [BC] (9.29) | 7.8 [A] (1.38) | 22.58 [C] (2.46) | 7.93 [A] (1.45) | <0.0001 * |
| Spine width (mm) | 1.8 [C] (0.63) | 0.82 [AB] (0.14) | 1.22 [BC] (0.32) | 0.76 [AB] (0.16) | 1.63 [C] (0.28) | 0.47 [A] (0.04) | <0.0001 * |
| **Anatomical (μm)** | | | | | | | |
| Stomata density | 30.18 [A] (4.83) | 44.97 [B] (17.45) | 30.31 [A] (5.43) | 26.41 [A] (2.22) | 28.08 [A] (4.23) | 35.32 [AB] (5.06) | 0.017 * |
| Cuticle thickness | 22.04 [B] (8.52) | 12.40 [AB] (2.62) | 21.44 [B] (2.31) | 21.28 [B] (7.93) | 20.72 [AB] (4.91) | 9.04 [A] (2.51) | 0.005 * |
| Epidermis thickness | 40.72 (15.15) | 30.67 (3.06) | 43.84 (3.44) | 33.44 (9.71) | 34.32 (7.26) | 28.00 (2.83) | 0.073 |
| Hypodermis thickness | 161.04 (57.37) | 143.60 (25.30) | 119.44 (8.78) | 153.76 (23.36) | 126.24 (15.72) | 147.60 (26.62) | 0.274 |
| **Biomass Partitioning (%)** | | | | | | | |
| Spine DW | 12.28 [C] (2.06) | 5.81 [AB] (4.05) | 4.55 [AB] (2.75) | 4.51 [AB] (3.87) | 10.02 [BC] (1.47) | 2.49 [A] (1.41) | <0.0001 * |
| Aboveground stem DW | 40.13 (12.19) | 26.9 (20.19) | 25.41 (13.72) | 33.53 (9.46) | 31.84 (4.71) | 45.85 (6.66) | 0.08 |
| Underground stem DW | 39.61 (10.08) | 56.73 (27.10) | 64.68 (13.53) | 54.15 (17.55) | 53.6 (5.98) | 38.44 (5.82) | 0.05 |
| Main root DW | 6.7 [AB] (2.68) | 8.91 [AB] (1.97) | 4.99 [AB] (1.11) | 7.11 [AB] (6.11) | 3.48 [A] (1.03) | 11.23 [B] (1.84) | 0.012 * |
| Secondary root DW | 1.28 [AB] (0.96) | 1.65 [AB] (1.40) | 0.38 [A] (0.28) | 0.69 [AB] (0.26) | 1.06 [AB] (0.74) | 1.98 [B] (0.58) | 0.027 * |

*G. hybopleurum* (Figure 2C) exhibited a long napiform root that represented almost 50% of its total length, an aboveground stem with intermediate density of areoles, and many large and wide spines, but fewer in comparison to *G. pugionacanthum* and *G. oenanthemum*. *G. stellatum* (Figure 2D) individuals also had the largest number of ribs of the studied species. Its underground stem was twice as long and the aboveground stem and presented a high density of areoles with a low number of tiny spines.

*G. oenanthemum* (Figure 2E) exhibited a large main root, which represented more than 50% of the total length of an individual, and an aboveground stem with a high number of long and wide spines, but a low density of areoles. *G. baldianum* (Figure 2F) also presented a long and wide main root, and the aboveground stem was shorter than the underground one. This species had the highest density of areoles of the studied species, with a low number of tiny spines.

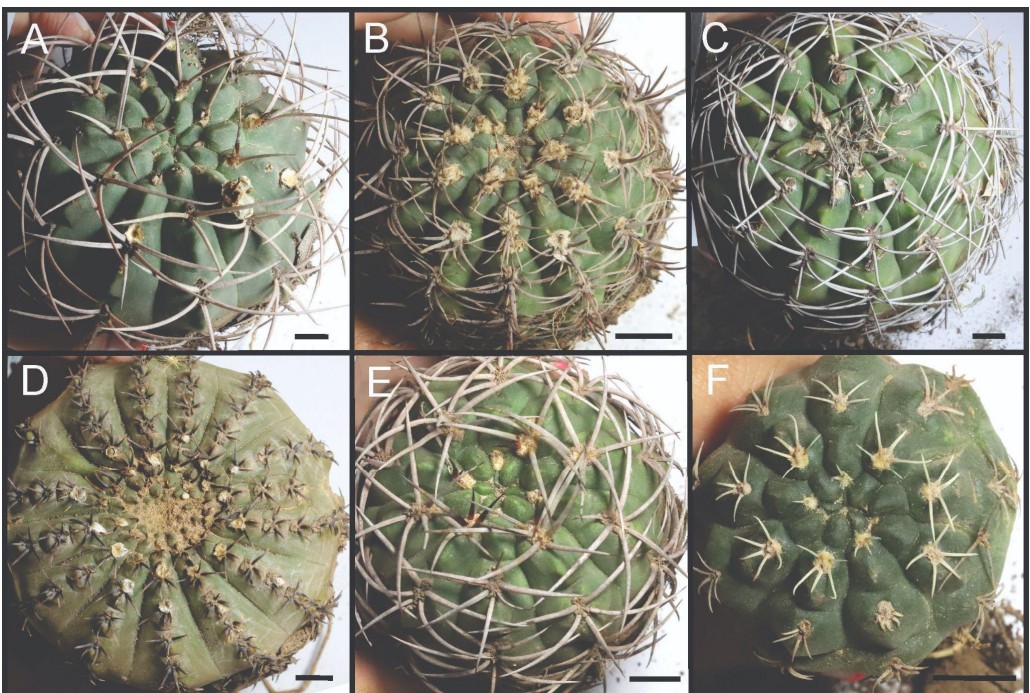

**Figure 2.** Top view of individuals of *Gymnocalycium* species. (**A**): *G. pugionacanthum* and (**B**): *G. marianae*, both from the arid site. (**C**): *G. hybopleurum* and (**D**): *G. stellatum*, both from the semiarid site. (**E**): *G. oenanthemum* and (**F**): *G. baldianum*, both from the subhumid site. Scale: 1 cm.

Only a few morphological variables showed a pattern linearly related to the aridity gradient. The analysis of the three species occurring along the whole gradient and belonging to the same subgenus (*G. pugionacanthum*, *G. hybopleurum,* and *G. oenanthemum* of Scabrosemineum) showed that the length of aboveground and underground stems varied with environmental characteristics, becoming shorter as humidity increased, whereas the length of the main root shows the inverse pattern: it decreased with increasing humidity. Similarly, in the species belonging to the Gymnocalycium subgenus (*G. marianae* and G. *baldianum*), as humidity increased, the length of the aboveground stem decreased, and the length of the underground stem and main root increased.

*3.2. Biomass Partitioning*

As shown in Table 1, *G. baldianum* allocated significantly less biomass to spines than the other species, whereas *G. pugionacanthum* allocated more biomass. *G. oenanthemum* presents significantly less biomass allocated to the main root than *G. baldianum*. Similarly, *G. baldianum* allocated significantly more biomass to the secondary and thin roots than *G. hybopleurum*. Regarding the biomass allocated to the stem, none of the analyzed species showed significant differences (Figure 3).

In all species, biomass allocation to underground stems was considerably higher than to aboveground stems, except for *G. pugionacanthum,* which allocated a similar amount of biomass to both parts of the stem, and *G. baldianum*, which allocated more biomass to the aboveground part of the stem. Moreover, in all species, more biomass was allocated to the main root than to the secondary and thin roots. This variable was the only one to show a linear pattern regarding the gradient. Indeed, *G. pugionacanthum*, a species that occurs at the arid extreme of the gradient, allocated more biomass to the main root. However, the analysis of the subgenus Gymnocalycium shows that *G. baldianum* allocated more biomass to the main root than *G. marianae*, even though the former occurs in more humid environments (Table 1). Therefore, although *G. marianae* inhabits more arid environments, biomass allocation to the main root is not as high as expected.

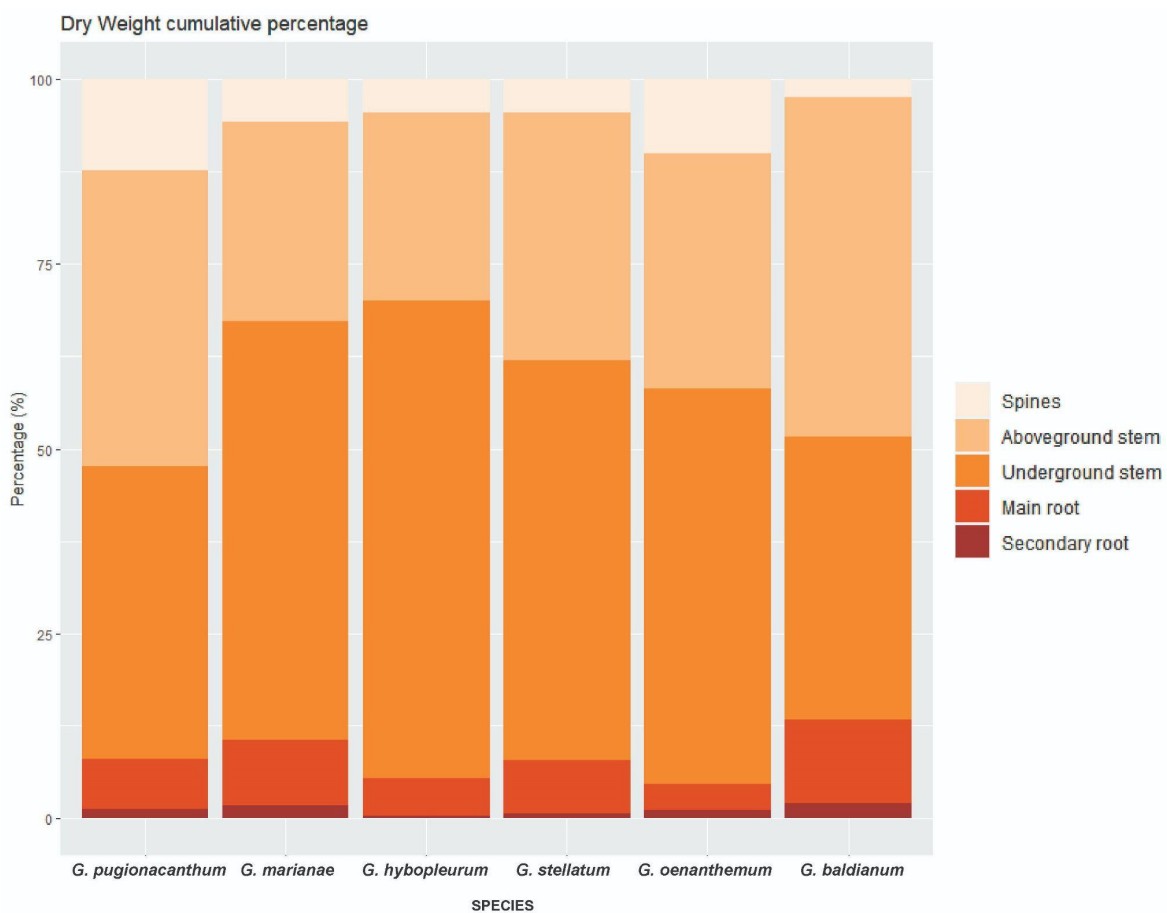

**Figure 3.** Biomass partitioning and allocation expressed as percentage of dry weight. Percentages correspond to each part of the plant. Species are ordered according to the precipitation gradient, from the arid (*G. pugionacanthum* and *G. marianae*) to the humid conditions (*G. oenanthemum* and *G. baldianum*).

### 3.3. Anatomical Traits

The studied species presented both qualitative and quantitative differences in the dermal system. Epidermis was the most variable tissue among species. In surface view, *G. pugionacanthum* (Figure 4A), *G. hybopleurum* (Figure 4B), *G. oenanthenum* (Figure 4C), and *G. stellatum* (Figure 4D) presented sunken stomata and an elliptical thickening or outer rim of cuticle. *G. pugionacanthum* and *G. stellatum* presented large pustules or papillae, which were smaller in *G. hybopleurum* and *G. oenanthemum*. In contrast, in *G. marianae* (Figure 4E) and *G. baldianum* (Figure 4F), stomata were at the same level as the epidermal cells and were very abundant (Table 1).

In cross section, the epidermis was unstratified, with a notable cuticle that varies in thickness (Table 1). In most species, epidermal cells had different sizes and shapes (Figure 4 D–G). *G. pugionacanthum* (Figure 4D), *G. hybopleurum* (Figure 4E), and *G. stellatum* (Figure 4G) presented large pustules formed by groups of 2–4 larger epidermal cells, covered with thick cuticle, while *G. oenanthenum* (Figure 4F) presented small undulations. However, in *G. marianae* (Figure 4H) and *G. baldianum* (Figure 4I), the epidermis and cuticle were homogeneous and smooth, and the epidermal cells had a similar shape and size, although some may be larger and form small undulations.

The substomatal chambers were deep and formed a cavity that traversed the entire width of the hypodermis, down to the underlying chlorenchyma, being notable in *G. pugionacanthum* (Figure 4D). The hypodermis in all species consisted of three to six strata of cells with strong walls. The thickenings were stained with fucsin basal and safranin, but not with phloroglucinol, so they are not lignified thickenings; further histochemical tests are necessary to corroborate their chemical nature. Development of strata varied according to the species (Table 1). The chlorenchyma was composed of palisade cells with chloroplasts, located perpendicular to the surface of the stem. In most species, the chlorenchyma was a compact, resistant tissue with some mucilage ducts (Figure 4D–G), whereas in *G. baldianum* and *G. marianae*, it was noticeably laxer and more fragile and with more abundant mucilage (Figure 4H,I).

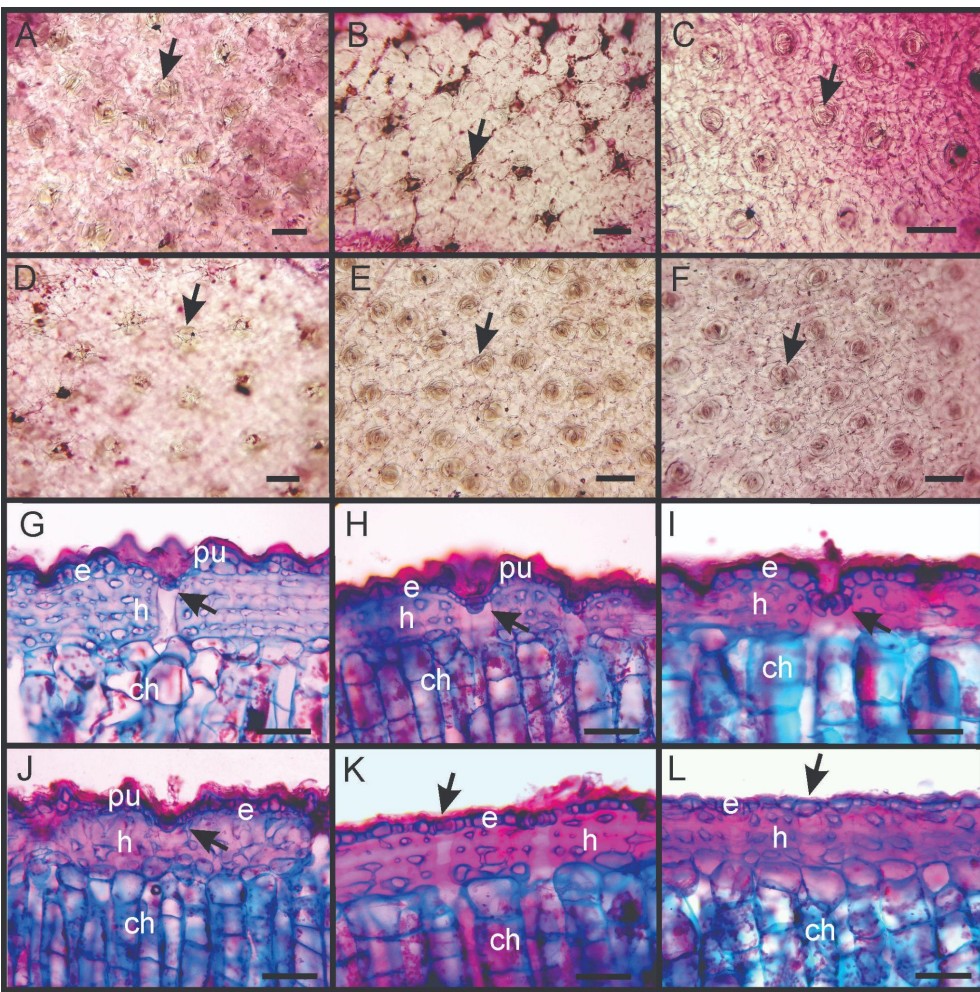

**Figure 4.** Stem anatomy in *Gymnocalycium* species. (**A–F**): Epidermis in surface view. Sunken stomata with cuticle ridges and pustules in (**A–D**). Stomata at the same level in (**E,F**). (**G–L**): Epidermal system and chlorenchyma in transverse section. Epidermis with pustules, irregular cuticle thickening and sunken stomata in (**G–J**). Thin and homogeneous epidermis and cuticle, stomata at the same level in (**H,I**). (**A,G**): *G. pugionacanthum*; (**B,H**): *G. hybopleurum*, (**C,I**): *G. oenanthenum* and (**D,J**): *G. stellatum*. (**E,K**): *G. marianae* and (**F,L**): *G. baldianum*. Scale: (**A–F**): 100 microns. (**G–L**):150 microns. Arrowhead: stomata. Abbreviations: ch: chlorenchyma, e: epidermis, h: hypodermis, pu: pustules.

Regarding the quantitative anatomical variables analyzed, only stomatal density and cuticle thickness showed significant differences among species. *G. marianae* and *G. baldianum* had more stomata, while *G. pugionacanthum*, *G. hybopleurum*, and *G. stellatum* had greater cuticle thickness; these three species also had thicker epidermis and hypodermis.

*Gymnocalycium pugionacanthum* presented the highest values in almost all the measurements. It exhibited thick cuticles, epidermis, and hypodermis, with these characteristics being in accordance with the dry environment where it occurs. The other species present intermediate values. On the other hand, *G. marianae* and *G. baldianum* are very similar to each other and different from the other species. They have the thinnest cuticle and epidermis, hypodermis of intermediate development, and abundant stomata.

### 3.4. Multivariate Analyses

We ran a principal component analysis (PCA) for 17 quantitative variables and 28 individuals (see Table A1). PC1 explained 27.1% of the total variation, whereas PC2 and PC3 explained 19.7% and 13.6%, respectively (Figure 5; Table 2). PC1 mainly explained the variation in spine length and width, and the density of areoles, followed by cuticle thickness and the dry weight of the main and secondary root, i.e., PC1 explained morphological variables and variables related to biomass partitioning. The dry weight of aboveground and underground stems and spines, as well as root length, are the variables that most contributed to the variation explained by PC2. Finally, the variables that were along PC3 were the length of above and underground stems, number of spines, and cuticle and hypodermis thickness. Overall, species dispersal in the Euclidean space was more related to their phylogeny than to the precipitation gradient since they were grouped according to their subgenera.

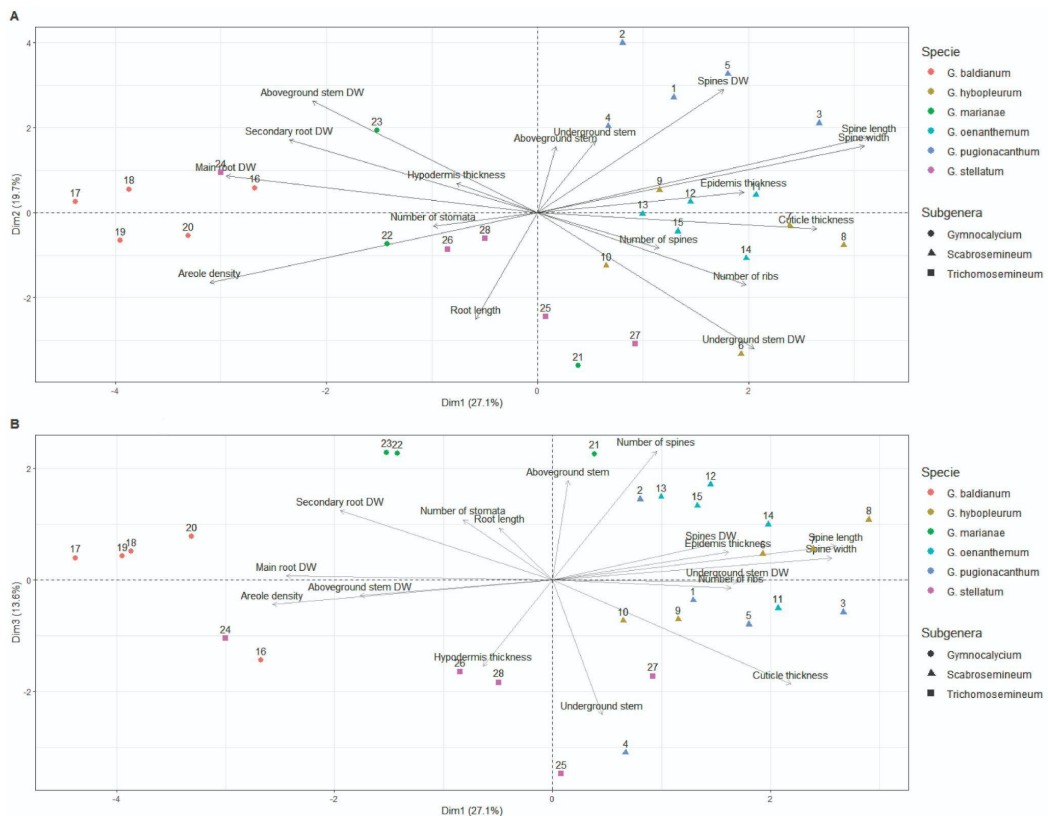

**Figure 5.** PCA biplots. (**A**): Representation of the first principal component (Dim 1) and the second one (Dim 2). (**B**): Representation of first and third principal components (Dim 3). The subgenera are represented with different shapes and the species with different colors.

**Table 2.** Eigenvalues, percentage of variance explained by each principal component, and cumulative percentage of variance. Notice that the first three components were considered, since they were the components that explained most of the variability.

|  | λ (Eigenvalue) | Variance (%) | Cumulative Variance (%) |
|---|---|---|---|
| PC1 | 4.61 | 27.11 | 27.11 |
| PC2 | 3.35 | 19.72 | 46.83 |
| PC3 | 2.31 | 13.60 | 60.43 |

## 4. Discussion

Our results highlight that species greatly differ in terms of their morphoanatomical traits and biomass partitioning [30]. Contrary to our expectations, the variations were mainly explained by phylogeny rather than by the characteristics of the environment where species occur. Species of the same subgenus were more similar among them, independently of the environment where they occur. It is remarkable that, even within a genus, phylogeny has an important role in determining species characteristics. Demaio et al. [31] showed that some traits evolved with the genus *Gymnocalycium*. For example, the subgenus *Gymnocalycium* has heavier seeds and a napiform root compared to the other two subgenera. This indicates phylogenetic constraints on functional attributes of a species, but further research is needed to understand how these constraints limit species distribution along environmental gradients.

Our study also shows that in a particular site, coexisting species can have different sets of trait attributes. The coexistence of species that differ in trait attributes is common in any community [32], but no study has analyzed this in globose cacti communities. Gurvich et al. [14] found that coexistence in a globose cactus community was partly related to microhabitat preferences of species. Functional differences of species that coexist in each site may be related to microenvironmental characteristics [14]. The possibility that, under particular climatic conditions, species with different strategies can coexist cannot be discarded.

Regarding the anatomical characteristics of the stems, in *Gymnocalycium* only the conduction and photosynthetic systems have been studied [33–36], so the anatomical descriptions presented are novel. Epidermal characteristics varied between *G. marianae* and *G. baldianun;* species of the subgenus Gymnocalycium show common family characteristics: uniseriate epidermis covered with a thin layer of cutin and abundant stomata at the same level [10,16–23]. However, the remaining species showed unusual characters (papilates and sunken stomata), which have been previously reported in a few genera, being important taxonomic features [17]. On the other hand, epidermis traits have an important physiological implication. Species with thick cuticle, sunken stomata, and a long substomatal chamber through the thick hypodermis, like *G. pugionacanthum* and *G. stellatum*, would have an extra advantage in arid environments. Indeed, these characteristics allow low water loss by transpiration, keeping the stomata open for gas exchange, and creating a $CO_2$ diffusion gradient between the chlorenchyma and the atmosphere [4,17,19,37].

The hypodermis in most cacti consists of collenchyma cells with thickened cellulose in their walls or with accumulations of pectic substances [17,18], but the thickenings are not lignified [38,39]. Our studies showed that the thickenings of the hypodermis are not cellulose or lignin, although they were stained with Safranin and Basic Fuchsin, so further studies are needed to corroborate the chemical nature and ultrastructure of this particular tissue. Its function is to contribute to the firmness of the dermal system [16] and is directly associated with the protection of the photosynthetic tissue from excess solar radiation in open vegetation [23,40]. The number of layers of the hypodermis and the cell wall thickness may be affected by environmental conditions [16] and can be related to the aridity and xeromorphy of the stems [39]; therefore, *G. pugionacanthum* and *G. stellatum* would be the best suited species for arid environments. In accordance with other traits analyzed in

the study, the anatomical characteristics of the species were better explained by species subgenera rather than by species distribution along the aridity gradient.

The analysis within a subgenus does also not show clear patterns in relation to the gradient. For example, in the subgenus Scabrosemineum, the only one with the presence of a species along the entire gradient, we found differences among species, but not always linearly related to the gradient. For example, aboveground stem DW was minimum in *G. hybopleurum* and maximum in *G. pugionacanthum* and *G. oenathenum*. We found linear patterns among the variables and the aridity gradient only in a few variables. When we analyzed the within-subgenus patterns, both in Scabroseminuem and Gymnocalycium, we found the same pattern only in two traits: cuticle thickness and length of aboveground stem (Table 1). In both traits, the values were higher in the arid site and lower in the humid one.

To our knowledge, this is the first study that analyzes intrageneric differences in globose cactus characteristics in an environmental context. Future work is needed to analyze whether the patterns found here hold true in other genera of globose cacti, and also in other life forms. Globose species represent an important part of total cactus richness [5]; therefore, the understanding of this group is important to predict their response to global change. Thirty percent of cactus species are in extinction risk due to human activities [41,42], and particularly due to global change [43,44]. The understanding of species characteristics and the factors that shape them could also be important to predict species responses to global changes.

**Author Contributions:** Conceptualization, D.E.G., S.B.P. and N.L.A.; methodology, S.B.P., N.L.A. and N.E.D. formal analysis, N.L.A. and S.B.P.; investigation, S.B.P., N.L.A., N.E.D., M.P. and A.H.; resources, A.H., M.P. and D.E.G.; data curation, S.B.P., N.L.A. and N.E.D.; writing—original draft preparation, S.B.P., N.L.A., N.E.D. and D.E.G.; writing—review and editing, D.E.G. and N.E.D.; visualization, S.B.P., N.L.A. and N.E.D.; supervision, D.E.G. All authors have read and agreed to the published version of the manuscript.

**Funding:** FONCyT N° 2016-0077 and 2017-0220. Research Committee of the Cactus and Succulent Society of America.

**Institutional Review Board Statement:** Not applicable.

**Informed Consent Statement:** Not applicable.

**Data Availability Statement:** Not applicable.

**Acknowledgments:** Jorgelina Brasca assisted with the English version of this manuscript. We are grateful to Marc Baker and two anonymous referees whose comments greatly improved the quality of this work.

**Conflicts of Interest:** The authors declare no conflict of interest.

## Appendix A

**Table A1.** Basic data matrix (BDM) for the principal component analysis (PCA): (1) aboveground stem (%); (2) underground stem (%); (3) root length (%); (4) number of ribs; (5) density of areola; (6) number of spines; (7) spine length (mm); (8) spine width (mm); (9) spine dry weight (%); (10) main root dry weight (%); (11) secondary root dry weight (%); (12) aboveground stem dry weight (%); (13) underground stem dry weight (%); (14) cuticle thickness (μm); (15) epidermis thickness (μm); (16) hypodermis thickness (μm); (17) number of stomata; (18) subgenus.

| Individual | 1 | 2 | 3 | 4 | 5 | 6 | 7 | 8 | 9 | 10 | 11 | 12 | 13 | 14 | 15 | 16 | 17 | 18 |
|---|---|---|---|---|---|---|---|---|---|---|---|---|---|---|---|---|---|---|
| *G. pugionacanthum* | 24.005 | 38.075 | 37.920 | 8 | 3 | 8 | 24.21 | 1.65 | 10.739 | 9.287 | 0.806 | 42.84 | 36.328 | 25 | 52.2 | 168.8 | 29.392 | Scabrosemineum |
| *G. pugionacanthum* | 38.766 | 30.053 | 31.181 | 11 | 3 | 5 | 25.17 | 1.37 | 13.997 | 5.587 | 2.837 | 47.78 | 29.8 | 14.6 | 55.6 | 122 | 24..128 | Scabrosemineum |
| *G. pugionacanthum* | 15.49 | 45.99 | 38.521 | 10 | 3.5 | 5 | 29.56 | 2.77 | 14.916 | 9.472 | 0.474 | 20.842 | 54.297 | 15.6 | 45 | 124.8 | 37.070 | Scabrosemineum |
| *G. pugionacanthum* | 19.97 | 42.222 | 37.807 | 10 | 3.5 | 5 | 25.36 | 1.17 | 10.25 | 6.003 | 1.54 | 36.969 | 45.237 | 35.4 | 19.2 | 258 | 32.244 | Scabrosemineum |
| *G. pugionacanthum* | 26.651 | 41.399 | 31.949 | 11 | 3 | 5 | 30.95 | 2.05 | 11.503 | 3.163 | 0.738 | 52.217 | 32.38 | 19.6 | 31.6 | 131.6 | 28.076 | Scabrosemineum |
| *G. hybopleurum* | 14.147 | 22.239 | 63.614 | 13 | 4.5 | 9 | 16.76 | 0.85 | 3.289 | 5.986 | 0.154 | 12.021 | 78.551 | 21.2 | 42.8 | 117.6 | 23.251 | Scabrosemineum |
| *G. hybopleurum* | 27.975 | 28.74 | 43.285 | 14 | 4 | 7 | 32.98 | 1..5 | 4.061 | 5.425 | 0.315 | 29.891 | 60.309 | 22 | 38.8 | 122 | 37.728 | Scabrosemineum |
| *G. hybopleurum* | 21.264 | 30.582 | 48.154 | 11 | 4 | 9 | 28.33 | 1.61 | 8.764 | 3.132 | 0.621 | 14.029 | 73.453 | 18.4 | 43.6 | 122.8 | 33.121 | Scabrosemineum |
| *G. hybopleurum* | 23.029 | 36.966 | 40.005 | 11 | 4 | 7 | 13.82 | 1.11 | 5.26 | 4.901 | 0.078 | 45.993 | 43.767 | 24.8 | 47.6 | 105.6 | 28.296 | Scabrosemineum |
| *G. hybopleurum* | 20.069 | 30.617 | 49.314 | 10 | 4.5 | 5 | 12.15 | 1.02 | 1.372 | 5.508 | 0.726 | 25.098 | 67.297 | 20.8 | 46.4 | 129.2 | 29.173 | Scabrosemineum |
| *G. oenanthemum* | 24.884 | 36.63 | 38.486 | 10 | 4 | 8 | 24.69 | 1.15 | 10.894 | 4.108 | 0.004 | 27.889 | 57.105 | 28.8 | 23.6 | 118.4 | 30.928 | Scabrosemineum |
| *G. oenanthemum* | 23.079 | 11.145 | 65.776 | 11 | 3 | 7 | 23.94 | 1.86 | 11.733 | 3.019 | 1.304 | 36.314 | 47.63 | 17.2 | 32.4 | 140 | 22.812 | Scabrosemineum |
| *G. oenanthemum* | 16.091 | 17.775 | 66.134 | 10 | 3 | 8 | 18.94 | 1.72 | 9.708 | 3.703 | 1.892 | 37.598 | 47.099 | 22 | 34 | 110.4 | 28.076 | Scabrosemineum |
| *G. oenanthemum* | 13.417 | 17.727 | 68.856 | 10 | 3 | 7 | 24.20 | 1.8 | 7.808 | 1.974 | 0.649 | 28.972 | 60.597 | 17.6 | 38.8 | 116.4 | 25.225 | Scabrosemineum |
| *G. oenanthemum* | 17.309 | 17.152 | 65.538 | 11 | 3 | 7 | 21.14 | 1.61 | 9.948 | 4.611 | 1.43 | 28.425 | 55.585 | 18 | 42.8 | 146 | 33.341 | Scabrosemineum |
| *G. baldianum* | 16.313 | 42.447 | 41.240 | 8 | 6 | 6 | 7.79 | 0.48 | 1.735 | 13.446 | 1.485 | 39.493 | 43.84 | 12 | 29.6 | 174 | 33.341 | Gymnocalycium |
| *G. baldianum* | 13.729 | 27.535 | 58.735 | 7 | 11 | 7 | 7.82 | 0.5 | 3.303 | 12.453 | 2.87 | 46.562 | 34.813 | 5.2 | 26.4 | 166.4 | 31.147 | Gymnocalycium |
| *G. baldianum* | 25.74 | 26.195 | 48.065 | 7 | 13 | 5 | 9.25 | 0.4 | 4.584 | 9.038 | 1.808 | 49.322 | 35.249 | 8.4 | 28.8 | 138 | 42.773 | Gymnocalycium |
| *G. baldianum* | 18.767 | 21.509 | 59.724 | 9 | 12 | 5 | 5.65 | 0.47 | 1.247 | 9.732 | 1.521 | 54.745 | 32.755 | 9.6 | 24 | 106.8 | 38.166 | Gymnocalycium |
| *G. baldianum* | 22.663 | 19.347 | 57.990 | 8 | 10 | 6 | 9.12 | 0.48 | 1.601 | 11.502 | 2.233 | 39.107 | 45.557 | 10 | 31.2 | 152.8 | 31.147 | Gymnocalycium |
| *G. marianae* | 32.981 | 19.137 | 47.881 | 12 | 10.5 | 11 | 11.52 | 0.66 | 1.178 | 6.68 | 0.256 | 6.011 | 85.876 | 15.2 | 34 | 115.6 | 41.018 | Gymnocalycium |
| *G. marianae* | 28.837 | 20.751 | 50.412 | 13 | 7.3 | 8 | 10.91 | 0.89 | 7.587 | 10.406 | 1.652 | 28.363 | 51.991 | 12 | 28 | 150.4 | 64.049 | Gymnocalycium |
| *G. marianae* | 37.084 | 18.772 | 44.144 | 12 | 4 | 8 | 15.14 | 0.91 | 8.675 | 9.654 | 3.051 | 46.311 | 32.309 | 10 | 30 | 164.8 | 29.831 | Gymnocalycium |
| *G. stellatum* | 16.587 | 35.572 | 47.841 | 10 | 8.5 | 4 | 6.07 | 0.53 | 10.346 | 17.832 | 1.015 | 40.461 | 30.346 | 10 | 23.2 | 115.2 | 23.909 | Trichosemineum |
| *G. stellatum* | 10.59 | 38.435 | 50.975 | 13 | 11 | 3 | 8.51 | 0.98 | 0.491 | 4.288 | 0.439 | 27.17 | 67.612 | 30.8 | 30 | 154.8 | 27.418 | Trichosemineum |
| *G. stellatum* | 17.313 | 26.863 | 55.825 | 12 | 9 | 3 | 7.77 | 0.76 | 3.883 | 5.298 | 0.567 | 42.38 | 47.871 | 17.6 | 48.8 | 176.8 | 28.076 | Trichosemineum |
| *G. stellatum* | 17.146 | 26.509 | 56.344 | 15 | 7 | 5 | 9.66 | 0.74 | 1.831 | 2.553 | 0.516 | 20.253 | 74.847 | 25.6 | 29.2 | 166.4 | 24.128 | Trichosemineum |
| *G. stellatum* | 16.281 | 33.393 | 50.326 | 12 | 7.5 | 4 | 6.99 | 0.8 | 5.991 | 5.584 | 0.928 | 37.407 | 50.09 | 22.4 | 36 | 155.6 | 28.515 | Trichosemineum |

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
