# Peer review of "Biomass Partitioning and Morphoanatomical Traits of Six Gymnocalycium (Cactaceae) Species Occurring along a Precipitation Gradient"

_diversity, doi:10.3390/d14090749_

Round 1
Author Response
Review of the manuscript Biomass partitioning and morphoanatomical traits of six Gymnocalycium (Cactaceae) species occurring along a precipitation gradient.
This manuscript present information about the morphology and anatomy of six species of Gymnocalycium that show a discrete distribution in a region which the authors divided in three sites with different precipitation regimens and belong to three subgenera. Their prediction was that differences (biomass and morphoanatomical characteristics) between species will be related to the precipitation regimens. No mention of the subgenera were introduced here. For each site they studied two species. One point that is not clear along the text is: Any of the six species is distributed in more than one precipitation regimen or are they microendemic?
Response, almost all Gymocalycium species present very small ranges, and also inhabit a limited range of environmental conditions. This is what allowed us to have this design. In the region studied, each species is only found in an ecological region defined by climate.
Your results differentiate the species; however, plants of the same species do not group together, as for G. hybopleurum, G. marianae and G. stellatum, species with an ample variation, figure 5. I suggest to test species limits and the three precipitation regimens through discriminant analyses to support your discussion and conclusions that is not the environment but the phylogeny.
Response, the species are well defined. The intraspecific variations found are due to the natural variability of the characters measured. In this sense, we do not believe that a discriminant analysis is necessary or useful to answer the questions posed.
More information on the phylogeny is necessary in this paper, probably in the introduction and discussion. Are the species of each pair selected for each subgenus sister taxa? As explain now in the method selection was based on the precipitation regimen. In the way the discussion is presented as well as the results is confusing because the limits of distribution for each species is not clearly stated. Does speciation events occurred under the specific precipitation regimen as here mentioned or are the soil features also involved?
Response, the selection of species was based on their distribution in sites with different levels of precipitation, and obviously on the species present in each site.
On the other hand, there is no information about how speciation events occurred and how climate influenced them. As stated by the reviewer, it would be very interesting to analyze the speciation processes in relation to climate and other environmental variables.
Other specific comments are given below
Line 52 it is mentioned that no typical wood is present. I agree that no leaf area can be calculated in Cactaceae, However, wood density it is possible to calculate, however it has been done in very few species hey have typical wood and wood density can be calculated see Romero E. et al. 2020. Wood density, deposits and mineral inclusions of successional tropical dry forest species. European Journal of Forest Research 139(3): 369-381 https://doi.org/10.1007/s10342-019-01236-9
Response, yes, in the case of columnar cacti where there is woody tissue it is possible to measure WD, as the work mentioned. But these globose species do not have woody tissue, so a measure of tissue density would not be comparable to WD.
Lines 72-73. The papers mentioned clearly do not do not test for aridity. However, there are other ones that allow us to know about the variability due to species distribution
Hernández M.Y. et al. 2007. Los estomas de Myrtillocactus geometrizans (Mart. ex. Pfeiff.) Console (Cactaceae): Variación en su área de distribución. Revista Fitotecnia Mexicana 30: 235-240.
Arias S. 2001. Variación en la anatomía de la madera de Pachycereus pecten-aboriginum (Cactaceae). Anales del Instituto de Biología, Universidad Nacional Autónoma de México, Serie Botánica 72(2): 157-169.
Aquino D et al., 2021. The importance of environmental conditions in maintaining lineage identity in Epithelantha (Cactaceae). Ecology and Evolution 11: 4520-4531. DOI: 10.1002/ece3.7347
Response, thank you for providing these articles. They were cited in the manuscript
Line 124 if the central spine was not the longest; which was the position of the “most promiscuous”. You mean that all spines in the areole were similar and no differences between central and radial was evident.
Response, Some species, such as G. baldianum and G. oenanthemum, have tiny spines with similar sizes with no clear pattern of distribution in the areola. Not always was evident which was the central spine so we chose the most prominent to be able to compare between species.
Line 137. Change “to perform preparations” for more proper words to perform temporary slides. Done
Lines 138-139. Add that they were 5 squares per plant per species Done
Line 140. Change “preparations” for slides Done
Line 141. Although Basin Fuchsin stains for lignin, the color in your photos does not clear support your results. Test those hypodermis with phloroglucinol to confirm your results since this a unique trait for the species studied of the genus not previously reported.
Response, Thank you for this suggestion, we performed the staining with phloroglucinol and it was negative, so the thickening is not lignin. We also stained again with Astral Blue, which was negative, so it is not cellulose thickening, and we repeated the positive stains for Fuchsin and Safranin.
Undoubtedly, the tests performed are not enough to corroborate the chemical nature of this particular tissue, therefore, in the results and in the discussion, we suggest the need for further studies, which are beyond the current capabilities of our laboratory.
Lines 142-143. What you mean by 10 samples. Are they 10 measurements per slide per plant per spp. If my interpretation is correct you have pseudoreplicates. How you analyzed this type of data in your univatiate Anova test.
Response, We calculated the mean value per individual previous to the statistical analysis.
Lines 146-147. Indicate the purpose of PCA, it was not required to transform variables; all fulfill the assumptions of the statistical multivariate analysis.
Response, The objective of the multivariate analysis was to analyze the morphofunctional and biomass particle characteristics in an integrated manner.
Reviewer 2 Report
-The ms is generally very well and clearly written.
-Line 98: “corresponds to?” Is it not there just corresponds to? Or do they just mean that it is IN that ecoregion? Also line 102.
-Line 121: how did you relativize the values? Turn them to proportions? Line 132 also.
-Line 124: sexual connotations aside, I am not sure that the word promiscuous is being used correctly here. Perhaps the authors mean prominent? Or a different word?
-Table 1 is hard to visually process in how it is laid out. Perhaps put the SD below each mean so as your eye looks across the line, you get related values. Do NOT add another horizontal line for SD, just put it (such as in parentheses) in the white space below the mean.
-I like how the components were individually described with regards to the grouping patterns. I find it frustrating that not all authors do this in their work.
-The font size of the key on the right side in Fig 5 could be enlarged. I like the use of color here.
-Table 2: given that about 40% of the variance is not explained here, I was wondering if the authors had any thoughts about what other factors may be at play here.
-I wonder if the authors had considered more species (e.g., not just globose cacti in one genus) then environmental influences would have been teased out. I like that phylogeny was included and controlled for, but I wonder if a larger sample generally would have yielded more results. It’s hard to draw really broad conclusions from a small sample. I don’t suggest the authors redo the work, but perhaps add a few lines in the Discussion about how a larger sample could have potentially yield different kinds of results. Even looking at the images in Figure 2, it was obvious that spines likely were not going to fall along a precipitation gradient, making me wish for more varied data to offer more insight (and thus not allowing one species to erase any hope of finding a potential underlying pattern). For example, if G. stellatum (Fig 2) is from an intermediate site, I think that alone was going to knock spines off of as a trend-setting variable. Following, I think the Discussion is also narrow/focused to follow suit with the spectrum of data used, which is completely appropriate, but that then also makes one thirst for broader explanation. The authors are right not to delve too deeply beyond what they have found and done, but perhaps some speculation about broader trends would be interesting, if they are able.
Author Response
-The ms is generally very well and clearly written.
-Line 98: “corresponds to?” Is it not there just corresponds to? Or do they just mean that it is IN that ecoregion? Also line 102.
Response, It means that those sites are in the mentioned ecorregion. Corrected.
-Line 121: how did you relativize the values? Turn them to proportions? Line 132 also.
Response, We turned them into percentages. All relativized values are detailed in appendix 1.
-Line 124: sexual connotations aside, I am not sure that the word promiscuous is being used correctly here. Perhaps the authors mean prominent? Or a different word?
Response, That is correct; we changed the word.
-Table 1 is hard to visually process in how it is laid out. Perhaps put the SD below each mean so as your eye looks across the line, you get related values. Do NOT add another horizontal line for SD, just put it (such as in parentheses) in the white space below the mean.
Response, Done
-I like how the components were individually described with regards to the grouping patterns. I find it frustrating that not all authors do this in their work.
Response, Thank you very much!
-The font size of the key on the right side in Fig 5 could be enlarged. I like the use of color here.
Response, Done
-Table 2: given that about 40% of the variance is not explained here, I was wondering if the authors had any thoughts about what other factors may be at play here.
Response, It is normal that in a multivariate analysis there is this percentage of the variance that is not explained by the axes. This simply depends on the variables chosen and the relationships that exist between them.
-I wonder if the authors had considered more species (e.g., not just globose cacti in one genus) then environmental influences would have been teased out. I like that phylogeny was included and controlled for, but I wonder if a larger sample generally would have yielded more results. It’s hard to draw really broad conclusions from a small sample. I don’t suggest the authors redo the work, but perhaps add a few lines in the Discussion about how a larger sample could have potentially yield different kinds of results. Even looking at the images in Figure 2, it was obvious that spines likely were not going to fall along a precipitation gradient, making me wish for more varied data to offer more insight (and thus not allowing one species to erase any hope of finding a potential underlying pattern). For example, if G. stellatum (Fig 2) is from an intermediate site, I think that alone was going to knock spines off of as a trend-setting variable. Following, I think the Discussion is also narrow/focused to follow suit with the spectrum of data used, which is completely appropriate, but that then also makes one thirst for broader explanation. The authors are right not to delve too deeply beyond what they have found and done, but perhaps some speculation about broader trends would be interesting, if they are able.
Response, This work is part of a doctoral thesis where more variables are being measured. It is very interesting what the reviewer remarked, and in fact it would be very interesting to have more species and of other life forms, but for the moment it escapes our objectives. We include a sentence in the last paragraph of the discussion stating the need for further studies in other genera of globose cacti and even in species of other life forms.
Reviewer 3 Report
I like your study and I think that more like it are needed. I realize that because of the destructive sampling that large sample sizes are not possible. Do you think that you could find a trait that might vary according to aridity that could be sampled without harming individuals? All in all, you did a good job. Some of the wording is a little awkward but generally clear.

Author Response
I like your study and I think that more like it are needed. I realize that because of the destructive sampling that large sample sizes are not possible. Do you think that you could find a trait that might vary according to aridity that could be sampled without harming individuals? All in all, you did a good job. Some of the wording is a little awkward but generally clear.
I am not seeing this (and G. baldianum, which allocated more biomass to the aboveground part of the stem) Linea 197
Response, this information is in Table 1.
I can not make sense of this
Response, This information is in table 1. Anyway, we clarify this point in the text.
Línea 239 : (Regarding the quantitative anatomical variables analyzed, only stomatal density and cuticle thickness showed significant differences among species) is this statistically different or simply markedly different?
Response, It is statically different between species (in table one are the p-values)
Round 2
Reviewer 1 Report
The authors have incorporated some the comments done to the first version of this manuscript. They responded to those not incorporated. I consider the manuscript is stronger. Only to minor typographic mistakes
Check line 234 the name of the staining
line 416 the reference is Arias, S. and T. Terrazas